# ALPHAFORMER: END-TO-END SYMBOLIC REGRESSION OF ALPHA FACTORS WITH TRANSFORMERS

## ABSTRACT

Identifying predictive patterns for stock market trends, known as alpha factors, is a critical challenge in quantitative finance. Symbolic regression (SR) methods can discover these factors as interpretable mathematical expressions, offering advantages over "black-box" machine learning approaches and manual methods that rely heavily on human expertise. However, existing SR methods typically restart the discovery process for each new dataset, failing to leverage prior knowledge. To address this limitation, we propose AlphaFormer, an encoder-decoder Transformer model designed for the end-to-end generation of synergistic alpha factors from raw stock market data. AlphaFormer leverages pre-training on synthetic datasets to efficiently uncover synergistic alpha factors for new datasets, capitalizing on acquired prior knowledge. To overcome the challenge of generating synthetic stock datasets with temporal dependencies, we introduce a novel generative framework that integrates multiple time-series generative models to generate synthetic stock data and dynamically select the highest quality samples, ensuring the creation of high-fidelity datasets crucial for pre-training. Extensive evaluations on real-world stock market datasets demonstrate that AlphaFormer outperforms existing methods across widely used metrics, achieving superior performance with significantly reduced inference computation—generating only 33% as many factors as the best baseline and requiring no further training during inference. Backtests further show that AlphaFormer delivers the highest annual return among all methods, highlighting its practical potential for superior investment performance.

## 1 INTRODUCTION

In the dynamic landscape of financial markets, achieving investment returns exceeding a benchmark is a primary objective for investors and financial institutions. Consistently generating this superior performance relies on the ability to identify predictive patterns in financial data Qian et al. (2007). These predictive patterns are referred to as alpha factors, serving as the fundamental building blocks for forecasting stock market trends Tulchinsky (2019). The systematic process of discovering these predictive factors is termed alpha mining. Therefore, identifying novel and exploitable alpha factors is a critical area of research within quantitative finance, serving as the foundation for competitive trading strategies.

Extensive efforts have been dedicated to the challenge of alpha mining, giving rise to various approaches. Generally, current approaches can be categorized into two main paradigms: machine learning-based methods and formulaic alpha methods. Machine learning techniques, including tree-based models like LightGBM Ke et al. (2017) and XGBoost Chen & Guestrin (2016), and deep learning models such as LSTM Hochreiter & Schmidhuber (1997), have been extensively employed due to their capacity in uncovering intricate and often non-linear patterns. More recently, advanced sequence modeling architectures such as Transformer-based models Xu et al. (2021) and state-space models (e.g., S5 Smith et al. (2023)) have been explored for financial time-series representation learning, offering powerful tools for capturing long-range dependencies. However, despite their strong predictive capacity, these approaches remain largely opaque, further reinforcing the appeal of interpretable formulaic alphas Yu et al. (2023). In contrast, formulaic alpha methods aim to discover explicit mathematical expressions as alpha factors—such as "mean(close, 20d)"—offering greater simplicity and interpretability. Valuing these attributes of transparency and analytical tractability, this work focuses on the second paradigm.

Research into formulaic alpha methods has historically evolved through two stages: manual approaches and symbolic regression (SR)-based methods. Initially, the field relied predominantly on manual approaches, where experts identified potential factors based on economic theory, intuition, or standard statistical analyses Kakushadze (2016). However, these methods were inherently limited by their dependence on human expertise and lacked the flexibility to systematically explore the vast space of possible factors. Recognizing the need for more systematic and data-driven discovery, the field advanced to SR-based methods, which encompass Genetic Programming (GP)-based and Reinforcement Learning (RL)-based approaches. GP-based methods employ evolutionary algorithms operating on expression trees to automatically search for and evolve factor formulas Cui et al. (2021); Zhang et al. (2020). More recently, RL-based SR methods have emerged Yu et al. (2023); Shi et al. (2025); Ren et al. (2024); Shin et al. (2024). These methods use a policy network, typically implemented with RNNs, to output an expression. The output expression obtains a reward based on some goodness-of-fit metric (e.g., Information Coefficient), allowing the policy network to learn to assign higher probabilities to more effective alpha factors. While these SR-based methods offer significant improvements in flexibility and automation over manual techniques, a limitation persists: both GP-based and RL-based SR methods often initiate the factor search anew for each distinct stock dataset, failing to leverage knowledge from prior discovery efforts.

Recent advancements in pre-trained SR provide a compelling solution to the limitation of traditional SR methods Biggio et al. (2021); Kamienny et al. (2022); d'Ascoli et al. (2023a;b); Becker et al. (2023); d'Ascoli et al. (2022). By utilizing an encoder-decoder transformer model Vaswani et al. (2017) pre-trained on large-scale synthetic datasets, pre-trained SR equips the model to generalize symbolic regression tasks. This enables it to directly infer mathematical expressions for new datasets by leveraging acquired prior knowledge. Inspired by this innovation, we propose **AlphaFormer**, an encoder-decoder Transformer model designed to generate synergistic symbolic expressions for alpha factors (e.g., "mean(close, 20d)") from a raw stock dataset in an end-to-end fashion. Our AlphaFormer can leverage prior knowledge gained from pre-training for efficient inference, without the extensive search required by traditional symbolic regression approaches.

A key challenge in pre-training AlphaFormer lies in the generation of synthetic stock datasets with temporal dependencies. Prior studies in pre-trained SR often relied on manual approaches to construct synthetic datasets from simple distributions like mixtures of Gaussians Biggio et al. (2021); Kamienny et al. (2022); Valipour et al. (2021). However, these methods are impractical for synthetic stock data generation due to the inherent temporal dependencies within stock data. To address this challenge, we introduce a novel generative framework specifically designed to create high-fidelity synthetic stock datasets for pre-training AlphaFormer. This framework generates synthetic stock data from multiple time-series generative models—including GRU Dey & Salem (2017), Transformer Vaswani et al. (2017), and diffusion models Tashiro et al. (2021)—and dynamically selects the synthetic data with the highest quality. This dynamic selection strategy, leveraging the complementary strengths of different generative models, ensures the creation of more realistic synthetic stock datasets, ultimately enhancing the alpha discovery capability of AlphaFormer.

Our contributions can be summarized as follows:

1. We propose AlphaFormer, an encoder-decoder Transformer model tailored to generate synergistic alpha factors from raw stock market data in an end-to-end manner.

2. We introduce a novel generative framework that integrates multiple time-series generative models to generate synthetic stock data and dynamically select the highest quality samples, ensuring the creation of high-fidelity datasets essential for pre-training AlphaFormer.

3. We extensively evaluate AlphaFormer on real-world stock market datasets, demonstrating superior performance over existing methods. Specifically, AlphaFormer surpasses existing baselines on the tested stock market datasets across widely used metrics, while generating only 33% as many factors as the best baseline method and requiring no further training during inference. Additionally, in a simulated trading environment, our approach yields the highest annual return compared with other baseline methods, highlighting its potential for superior investment returns in practice.

## 2 PRELIMINARY

### 2.1 ALPHA FACTOR

We consider a stock dataset $\mathcal{D} \in \mathbb{R}^{S \times K \times T}$ comprising $S$ stocks over a total of $T$ trading days with $K$ features (e.g., opening and closing prices). This dataset is a three-dimensional tensor where the dimensions represent stocks, features, and time, respectively. On each trading day $t \in \{1, 2, \ldots, T\}$, each stock $i \in \{1, 2, \ldots, S\}$ is associated with a feature matrix $x_{ti} \in \mathbb{R}^{K \times t}$, which contains the $K$ features over the past $t$ days. Let $X_t \in \mathbb{R}^{S \times K \times t}$ denote the tensor whose $i$-th row is $x_{ti}$. An alpha factor is defined as a function $f : \mathbb{R}^{S \times K \times t} \to \mathbb{R}^S$ that maps the feature tensor $X_t$ to a vector of alpha values $z_t = f(X_t) \in \mathbb{R}^S$, where each component corresponds to a stock.

### 2.2 ALPHA MINING

The objective of alpha mining is to identify a set of up to $M$ synergistic alpha factors capable of forecasting stock trends, specifically the future returns of all stocks, denoted by $y_t \in \mathbb{R}^S$ for each trading day $t$.[1] This collection of alpha factors, limited to a maximum size of $M$, is termed the Alpha Pool. We combine the factors within the Alpha Pool using a linear model. Given $m \leq M$ alpha factors $f_1, f_2, \ldots, f_m$, the composite alpha vector for day $t$ is expressed as:

$$z_t = g(X_t) = \sum_{k=1}^{m} w_k f_k(X_t), \tag{1}$$

where $w = (w_1, w_2, \ldots, w_m)$ represents the weights assigned to each factor. These weights $w$ are optimized by minimizing the following loss function:

$$\mathcal{L}(w) = \frac{1}{ST} \sum_{t=1}^{T} \|g(X_t) - y_t\|_2^2 + \lambda \|w\|_1, \tag{2}$$

where $\lambda$ is the regularization coefficient, the first term measures the mean squared error between the predicted and actual returns, and the second term promotes sparsity in the weights.

We evaluate forecasting performance using the average Information Coefficient (IC) and the average Rank IC. The daily IC for day $t$ is the Pearson correlation between the predicted alpha vector $g(X_t)$ and the target return vector $y_t$, denoted as $\sigma(g(X_t), y_t)$. The average IC is:

$$\bar{\sigma}(g(X), y) = \frac{1}{T} \sum_{t=1}^{T} \sigma(g(X_t), y_t). \tag{3}$$

Similarly, the daily Rank IC is the Pearson correlation between the ranks of $g(X_t)$ and $y_t$, denoted as $\sigma_{\text{rank}}(g(X_t), y_t) = \sigma(r(g(X_t)), r(y_t))$, and the average Rank IC is:

$$\bar{\sigma}_{\text{rank}}(g(X), y) = \frac{1}{T} \sum_{t=1}^{T} \sigma_{\text{rank}}(g(X_t), y_t). \tag{4}$$

### 2.3 FORMULAIC ALPHA

Formulaic alphas are alpha factors represented by mathematical expressions constructed from a predefined set of operators, features, and constants. The operators in these expressions fall into two categories: (i) elementary functions that operate on single-day data, such as addition $(+)$ and logarithm (log), and (ii) time-series operators that process data across multiple days, such as Max(close,10d), which computes the highest closing price of a stock over the most recent 10 days. Features are derived from raw stock data, including opening and closing prices, trading volumes, and other financial indicators, while constants are numerical values, such as the time window in a time-series operator (e.g., 10 or 20 days) or numeric literals in elementary functions (e.g., 0.5 or -1.0). A comprehensive list of all operators, features, and constants utilized in this framework is provided in Appendix B.

---

[1] A common definition of the future return is $(P_{s,t+N}/P_{s,t}) - 1$ for an $N$-day horizon, where $P_{s,t}$ is the closing price of stock $s$ on trading day $t$.

Figure 1: Framework for generating synthetic stock data with time dependencies. Multiple generative models produce candidate synthetic stock data from historical inputs. An LSTM Evaluator assesses each candidate and selects the data with the highest likelihood for inclusion in the synthetic dataset.

Each formulaic alpha is uniquely represented in Reverse Polish Notation (RPN), making the generation of an expression equivalent to generating its corresponding RPN sequence. For example, generating expression "mean(volume, 20d)" is equivalent to generating sequence [volume, 20d, mean, end], where end denotes the termination of the sequence. This equivalence simplifies the generation of alpha factors into a sequence generation task.

## 3 METHODS

### 3.1 DATA GENERATION

To pre-train a transformer-based model for generating alpha factors from a raw stock dataset end-to-end, we follow the practice in prior studies on pre-trained SR that utilized synthetic datasets $\{\mathcal{D}_i\}_{i=1}^N$ for pre-training Biggio et al. (2021); Kamienny et al. (2022); Valipour et al. (2021). In those studies, each synthetic dataset is limited to two dimensions–sample and feature–with samples treated as independent. This sample independence enabled researchers to manually generate samples from simple distributions to construct the dataset. In contrast, our synthetic stock dataset incorporates three dimensions: stock, feature, and time. The addition of the time dimension introduces temporal dependencies among stock features, rendering previous manual generation approaches impractical. To overcome this challenge, we propose a novel generative framework that integrates multiple time-series generative models, such as RNN-based and diffusion-based models, to capture the temporal dependencies of stock features effectively. The following paragraph details the generation process for our synthetic stock datasets.

For each synthetic dataset $\mathcal{D}_i$, we first sample the number of stocks $S$ uniformly from the range $[S_{min}, S_{max}]$, and the number of time steps $T$ uniformly from $[T_{min}, T_{max}]$. Then, for each stock in $\mathcal{D}_i$, we apply a dynamic selection strategy to select the highest-quality generated stock data from various generative models based on their likelihood, which serves as a measure of how well the generated sequences capture the temporal patterns observed in real-world stock data. As illustrated in Figure 1, this process proceeds as follows:

1. **Context Sampling**: A segment of real-world stock data with a fixed context window of size $c$ is sampled to serve as historical conditioning data.

2. **Data Generation**: Multiple generative models (e.g., GRU, Transformer, Diffusion-based) produce future stock data of length $T$, conditioned on the sampled historical stock data.

3. **Evaluation**: An LSTM-based evaluator estimates the likelihood of each generated future stock data sequence by computing the product of probabilities for each future time step.

4. **Selection**: The future stock data with the highest likelihood is selected for inclusion in the synthetic dataset, while all other sequences are discarded.

Our dynamic selection strategy leverages the complementary strengths of the generative models. For example, the RNN-based model excels at generating short time series, while the diffusion-

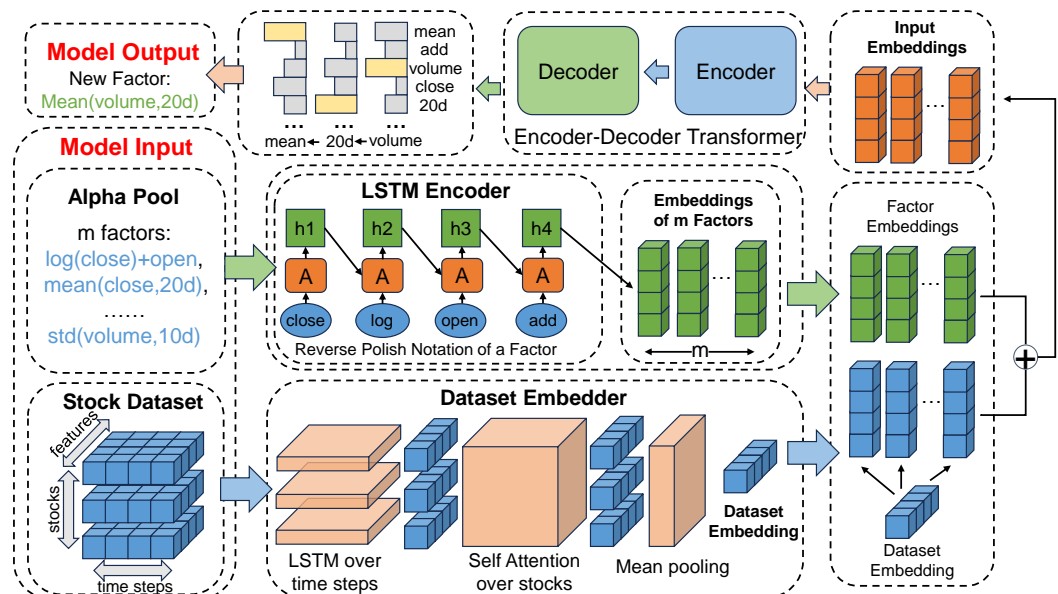

Figure 2: Architecture of our proposed alpha factor generator $P_\theta(f|\mathcal{D}, \mathcal{P})$, which is used to iteratively update the pool $\mathcal{P}$ of synergistic alpha factors by generating one new factor at a time. The generator takes the current Alpha Pool $\mathcal{P}$ and stock dataset $\mathcal{D}$ as inputs, producing a new alpha factor $f_{\text{new}}$ that is subsequently used to update the Alpha Pool.

based model is better suited for longer sequences Zhang et al. (2024). This adaptability ensures the generation of high-quality synthetic stock data for subsequent pre-training.

Both generative models and the LSTM evaluator are trained on stock data from the Chinese stock market, by optimizing the log likelihood or variational lower bound (for diffusion models). Detailed hyperparameters and pseudo-code of generating synthetic datasets are provided in Appendix C.

## 3.2 MODEL ARCHITECTURE

Our goal is to discover a *synergistic* set of alpha factors—i.e., factors whose joint predictive power exceeds the sum of their standalone effects because they capture complementary signals with limited redundancy and low collinearity—capable of forecasting stock trends within a given universe. We refer to this fixed-size collection of $M$ factors as the *Alpha Pool*. Constructing such a pool in one shot is computationally prohibitive: if the search space for a single factor is $L$, then the space for an $M$-factor pool scales as $L^M$. To make the problem tractable, we adopt an iterative procedure that updates the pool one factor at a time. At each step, we generate a candidate factor and evaluate its *conditional* contribution given the current pool—prioritizing incremental predictive gain and diversity—before integrating it into the Alpha Pool. The following paragraphs describe how new factors are produced and incorporated under this criterion.

The core of our iterative methodology is the alpha factor generator, $P_\theta(f|\mathcal{D}, \mathcal{P})$. This generator takes the current Alpha Pool $\mathcal{P}$ (containing $m$ factors) and a stock dataset $\mathcal{D} \in \mathbb{R}^{S \times K \times T}$ as inputs, and outputs a new alpha factor to refine the pool. As depicted in Figure 2, the generation process begins by transforming the dataset and existing factors into embeddings, as detailed below.

1. **Dataset Embedding**: The dataset embedder first processes each stock's data $\mathbf{s}_i \in \mathbb{R}^{T \times K}$ using a LSTM network. The hidden state of the LSTM's final time step for each stock $i$ serves as its stock embedding $\mathbf{h}_i \in \mathbb{R}^{n_{hid}}$ (where $n_{hid}$ is the dimension of the hidden state). These stock embeddings, $\{\mathbf{h}_i\}_{i=1}^{S}$, are then fed into a Transformer encoder. This encoder applies self-attention mechanisms to capture inter-stock relationships, producing contextual stock embeddings. Finally, a mean-pooling operation is applied across these contextual embeddings over the stock dimension to yield a dataset embedding $\mathbf{d} \in \mathbb{R}^{n_{hid}}$.

---

**Algorithm 1** Iterative refinement of an Alpha Pool

---

1: **Input:** An Alpha Pool $\mathcal{P} = \{f_j\}_{j=1}^m$, maximum pool size $M$, and new alpha factor $f_{\text{new}}$
2: **Output:** An updated Alpha Pool $\mathcal{P}^\star$
3: $\mathcal{P}' \leftarrow \mathcal{P} \cup \{f_{\text{new}}\}$
4: $w \leftarrow \arg\min_{w \in \mathbb{R}^{|\mathcal{P}'|}} \mathcal{L}(w)$              ▷ $\mathcal{L}(w)$ is defined by Equation (2)
5: **if** $|\mathcal{P}'| \leq M$ **then**
6:      $\mathcal{P}^\star \leftarrow \mathcal{P}'$
7: **else**
8:      $k \leftarrow \arg\min_{j=1,\ldots,|\mathcal{P}'|} |w_j|$          ▷ Find index of factor with smallest absolute weight
9:      $\mathcal{P}^\star \leftarrow \mathcal{P}' \setminus \{f_k\}$

---

2. **Factor Embeddings**: The $m$ alpha factors currently in the Alpha Pool are first converted into their Reverse Polish Notation (RPN) representation. Each RPN is a sequence composed of operators, input features, and constants. These RPN sequences are then processed by an LSTM encoder, where the last hidden state for each RPN sequence forms its corresponding factor embedding, resulting in $m$ factor embeddings $\{\mathbf{f}_i | \mathbf{f}_i \in \mathbb{R}^{n_{hid}}\}_{i=1}^m$.

Once the dataset embedding $\mathbf{d}$ and factor embeddings $\{\mathbf{f}_i\}_{i=1}^m$ are obtained, they are combined to create input embeddings $\{\mathbf{x}_i\}_{i=1}^m$ for generating a new factor: The dataset embedding $\mathbf{d}$ is replicated $m$ times, and these copies are added element-wise to the $m$ factor embeddings to obtain input embeddings, that is, $\mathbf{x}_i = \mathbf{d} + \mathbf{f}_i$. If the Alpha Pool is empty ($m = 0$), only the dataset embedding $\mathbf{d}$ is used as the input embedding. These combined input embeddings are then passed to an encoder-decoder Transformer model, which autoregressively generates the RPN sequence of a new alpha factor. This generated RPN is subsequently converted into its corresponding alpha factor expression.

Upon generation, the new alpha factor is integrated into the Alpha Pool. The newly generated alpha factor is provisionally added to the Alpha Pool. A linear model is then utilized to combine all factors within this augmented pool. The weights for this linear combination are determined by minimizing a loss specified in Equation (2). If the number of alpha factors ($m + 1$) in the Alpha Pool does not exceed the maximum size $M$, all factors and their corresponding weights are retained. However, if $m + 1 > M$, the alpha factor associated with the smallest absolute weight in the linear model is removed from the pool. This iterative procedure, as shown in Algorithm 1, allows for the incremental refinement of the Alpha Pool while strictly adhering to the size constraint $M$. Detailed hyperparameters of the model architecture are provided in Appendix D.

### 3.3 PRE-TRAINING AND INFERENCE

To identify synergistic alpha factors for predicting future stock returns, we optimize the IC of the Alpha Pool (Equation 3) as the objective for pre-training our generator $P_\theta(f|\mathcal{D}, \mathcal{P})$. During pre-training, for each stock dataset $\mathcal{D}$, the model performs num_iter iterations, generating one new alpha factor in each iteration to refine the Alpha Pool. In each iteration $j$, the model takes the dataset $\mathcal{D}$ and the existing Alpha Pool $\{f_i\}_{i=1}^m$, generates a new alpha factor $f_j$, and updates the Alpha Pool accordingly. The IC of the updated pool is computed and used as the reward $r_j$. We collect the tuple $(\mathcal{D}, \{f_i\}_{i=1}^m, f_j, r_j)$ from each iteration and use these tuples to update the model's parameters via the PPO algorithm Schulman et al. (2017), a stable and widely adopted reinforcement learning algorithm. Detailed hyperparameters and the pseudo-code of the pre-training process are provided in Appendix E.

During inference, with model parameters fixed, we initialize an empty Alpha Pool and run for Num Factors refinement steps, typically far exceeding the pool size $M$ (e.g., Num Factors = 20,000 while $M = 100$). At each step, the generator proposes a new candidate alpha factor conditioned on the dataset $\mathcal{D}$ and the current pool; the candidate is then evaluated and the pool is updated via Algorithm 1, retaining at most the top $M$ factors under our selection criterion.

## 4 EXPERIMENTS

We conduct comprehensive experiments to evaluate the performance of AlphaFormer. Our experimental design is structured to achieve four primary objectives: (i) to demonstrate AlphaFormer's ability to discover effective alpha factors with significantly reduced inference computation; (ii) to validate the effectiveness of our multi-model generative framework in enhancing AlphaFormer's alpha factor discovery; (iii) to assess its generalization capability across diverse stock markets; (iv) to confirm the practical effectiveness of AlphaFormer's generated factors under realistic trading conditions.

### 4.1 EXPERIMENTAL SETTINGS

**Datasets.** The experiments utilize daily raw market data from the Chinese A-shares market. Six fundamental features are selected as inputs for the alpha factors: {open, close, high, low, volume, and vwap (volume-weighted average price)}. The target is the 20-day holding period return, calculated as $\frac{\text{Ref(close}, -20)}{\text{close}} - 1$, where Ref(close, -20) denotes the closing price 20 days in the future. The dataset is chronologically divided into a training period (2004-01-01, 2018-12-31), and a testing period (2019-01-01, 2023-12-31). Our experiments focus on the constituent stocks of the CSI300 and CSI500 indices, which are widely used benchmarks in prior studies Yu et al. (2023); Shi et al. (2025).

**Baselines.** To comprehensively evaluate AlphaFormer, we benchmark it against methods from two primary paradigms, Detailed Hyperparameters of these baselines can be found in Appendix G:

1. **Machine Learning-based Methods**: These methods directly predict stock trends without generating explicit alpha factor expressions. They receive 60 days of raw features as input and are trained to predict the 20-day returns directly.

   **MLP:** A feedforward neural network modeling complex relationships in input features.
   **LSTM:** A Recurrent neural network capturing temporal relationships in input features.
   **XGBoost:** An efficient gradient boosting algorithm.
   **LightGBM:** A scalable gradient boosting framework.

2. **Symbolic Regression-based Methods**: These methods generate explicit mathematical expressions for alpha factors.

   **GP:** An evolutionary algorithm-based approach evolving mathematical expressions.
   **AlphaGen:** A reinforcement learning framework identifying synergistic alpha factors.
   **AlphaForge:** A generative predictive neural network for alpha factor generation.

**Evaluation Metrics.** We assess model performance using the following metrics:

- **IC**: Average information coefficient (see Eq. 3).
- **Rank IC**: Average rank information coefficient (see Eq. 4).
- **Num Factors**: For SR-based methods, the total number of generated alpha expressions during inference. This is **not** the Alpha Pool size (the pool is fixed at $M = 100$ for all SR methods). Num Factors serves as a proxy for inference-time computational effort.
- **Sharpe Ratio (SR)**: Risk-adjusted return, computed as mean return divided by return volatility.
- **Annual Return (CAGR)**: Annualized compounded growth rate over the evaluation horizon.
- **Maximum Drawdown (MDD)**: Worst peak-to-trough decline of the cumulative value (equity) curve.

### 4.2 MAIN RESULTS

To demonstrate AlphaFormer's superior alpha factor discovery, we compare AlphaFormer against the aforementioned baselines on both the CSI300 and CSI500 stock datasets.

As detailed in Table 1 and 2, AlphaFormer consistently achieves the highest IC and Rank IC across both datasets. This indicates its advanced stock selection ability compared to existing methods. Furthermore, AlphaFormer demonstrates this superior performance while generating only 33% as many factors as the best-performing baseline method. It is also crucial to note that AlphaFormer's

Table 1: Performance on CSI 300. Values are averaged over five random seeds ([0,1,2,3,4]); standard deviations in parentheses. `N/A` in `Num Factors` indicates the method does not generate factors.

| Methods | IC(%) ↑ | RankIC(%) ↑ | Num Factors ↓ | Sharpe Ratio ↑ | CAGR ↑ | MDD ↓ |
|---|---|---|---|---|---|---|
| MLP | 2.11 (0.33) | 2.81 (0.33) | N/A | 0.75 (0.09) | 0.13 (0.02) | -0.26 (0.03) |
| LSTM | 3.02 (0.66) | 3.49 (0.92) | N/A | 0.77 (0.08) | 0.13 (0.02) | -0.24 (0.04) |
| XGB | 3.55 (0.08) | 5.15 (0.13) | N/A | 0.83 (0.07) | 0.14 (0.01) | -0.22 (0.04) |
| LGBM | 3.86 (0.14) | 4.92 (0.12) | N/A | 0.80 (0.06) | 0.12 (0.01) | **-0.21** (0.02) |
| GP | 3.49 (1.14) | 4.00 (1.06) | 40000 | 0.67 (0.12) | 0.12 (0.02) | -0.27 (0.01) |
| AlphaGen | 5.19 (0.70) | 6.35 (0.60) | 60000 | 0.76 (0.07) | 0.13 (0.01) | -0.23 (0.04) |
| AlphaForge | 2.68 (0.80) | 4.42 (0.74) | 53461 | 0.47 (0.05) | 0.10 (0.01) | -0.44 (0.06) |
| **Ours** | **6.01** (0.32) | **6.95** (0.23) | **20000** | **0.86** (0.08) | **0.15** (0.02) | -0.23 (0.04) |

Table 2: Performance on CSI 500. Values are averaged over five random seeds ([0,1,2,3,4]); standard deviations in parentheses. `N/A` in `Num Factors` indicates the method does not generate factors.

| Methods | IC(%) ↑ | RankIC(%) ↑ | Num Factors ↓ | Sharpe Ratio ↑ | CAGR ↑ | MDD ↓ |
|---|---|---|---|---|---|---|
| MLP | 3.13 (0.28) | 4.99 (0.47) | N/A | 0.74 (0.07) | 0.15 (0.01) | -0.31 (0.02) |
| LSTM | 3.51 (0.53) | 5.31 (0.62) | N/A | 0.75 (0.09) | 0.16 (0.02) | -0.28 (0.04) |
| XGB | 3.04 (0.08) | 5.38 (0.11) | N/A | 0.71 (0.07) | 0.13 (0.01) | -0.27 (0.02) |
| LGBM | 3.99 (0.08) | 5.49 (0.12) | N/A | 0.85 (0.04) | 0.17 (0.01) | **-0.26** (0.02) |
| GP | 3.30 (1.16) | 4.21 (0.22) | 40000 | 0.53 (0.07) | 0.10 (0.01) | -0.33 (0.02) |
| AlphaGen | 3.48 (0.60) | 4.66 (0.27) | 60000 | 0.65 (0.07) | 0.12 (0.01) | -0.31 (0.02) |
| AlphaForge | 2.36 (0.68) | 5.64 (1.12) | 78362 | 0.21 (0.12) | 0.05 (0.03) | -0.45 (0.06) |
| **Ours** | **5.34** (0.29) | **5.82** (0.34) | **20000** | **0.91** (0.07) | **0.17** (0.01) | -0.29 (0.04) |

pre-trained model parameters remain fixed during the factor generation process, whereas methods like AlphaGen and AlphaForge require ongoing parameter updates. This distinction underscores AlphaFormer's enhanced computational efficiency at inference time. Example of the Alpha Pool genrated from our framework can be found in Appendix F.

### 4.3 IMPACT OF MULTI-MODEL SYNTHETIC DATA GENERATION FRAMEWORK

To validate the effectiveness of our multi-model generative framework, we conduct an ablation study. We pretrain separate AlphaFormer models using synthetic stock datasets generated by: (i) Our proposed multi-model generative framework (integrating GRU, Transformer, and diffusion models). (ii) Single-model generative frameworks (using GRU-only, Transformer-only, or Diffusion-only). These differently pre-trained AlphaFormer models are then evaluated on the CSI300 dataset. The performance comparison is presented in Figure 3.

Figure 3 shows that the AlphaFormer model pre-trained with synthetic datasets from our multi-model generative framework achieves the highest IC. This result validates our hypothesis that integrating the complementary strengths of multiple time-series generative models leads to a more effective pre-training phase for AlphaFormer, ultimately enhancing its alpha discovery capabilities.

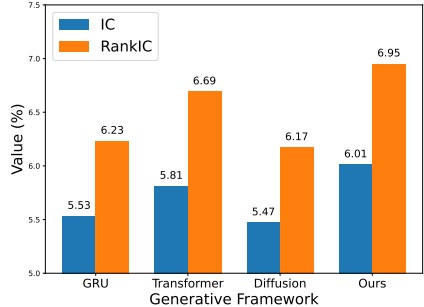

Figure 3: Performance comparison of AlphaFormer on the CSI300 dataset when pre-trained with synthetic datasets generated by different generative frameworks, which presents an ablation study on their impact.

### 4.4 GENERALIZATION ABILITY OF ALPHAFORMER

To evaluate the generalization ability of AlphaFormer, we apply AlphaFormer to the SP500 dataset, which comprises American stock data. Notably, the generative models used to create the synthetic

Table 3: Comparison of IC for different alpha mining methods on the SP500 dataset, which evaluates AlphaFormer's generalization ability. AlphaFormer is applied using its pre-trained model without further training on SP500 data, while baseline methods are trained on the SP500 dataset.

|  | MLP | LSTM | XGB | LGBM | GP | AlphaGen | AlphaForge | Ours |
|---|---|---|---|---|---|---|---|---|
| **SR** | 0.67 (0.11) | 0.72 (0.12) | 0.79 (0.10) | 0.79 (0.10) | 0.78 (0.13) | 0.79 (0.04) | 0.76 (0.16) | **0.80** (0.17) |
| **CAGR** | 0.10 (0.01) | 0.11 (0.02) | 0.12 (0.01) | **0.13** (0.02) | 0.11 (0.01) | 0.12 (0.01) | **0.13** (0.03) | 0.12 (0.02) |
| **MDD** | -0.27 (0.04) | -0.26 (0.03) | -0.25 (0.02) | -0.30 (0.01) | -0.27 (0.03) | -0.27 (0.02) | -0.25 (0.02) | **-0.24** (0.02) |

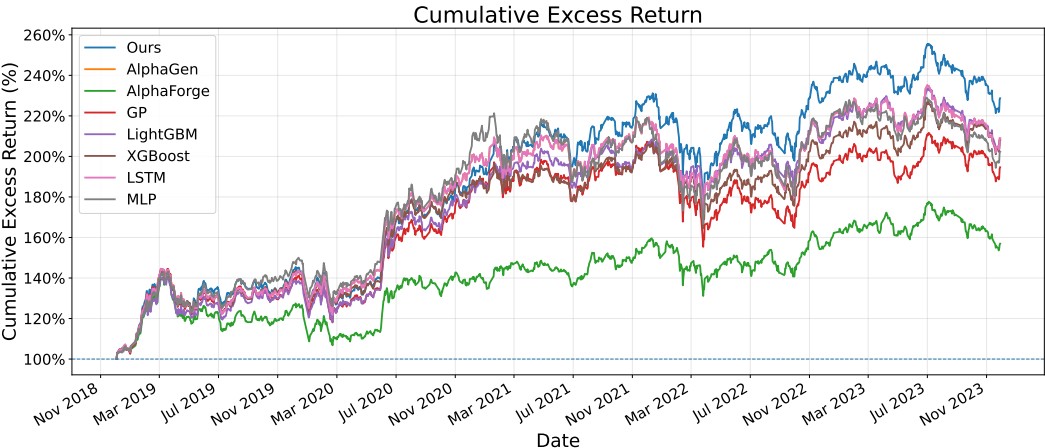

Figure 4: Backtest results on CSI300. We track the cumulative return of simulated trading agents utilizing the various alpha mining methods, which assess the practical utility of these methods.

datasets for pre-training AlphaFormer were trained on Chinese stock data. This setup tests AlphaFormer's ability to generalize across markets without additional adaptation to the SP500 dataset. In contrast, baseline methods are trained on the SP500 dataset following their standard procedures.

The performance of AlphaFormer and the baseline methods in the SP500 dataset is presented in Table 3. Despite being pre-trained on synthetic data from a different market, AlphaFormer demonstrates competitive performance on SP500 data, on par with baselines trained directly on that market. This result provides compelling evidence of AlphaFormer's robust cross-market generalization capability, showcasing its ability to generate effective alpha factors for an unseen market.

### 4.5 Performance in Simulated Trading Environment

To assess the practical utility of AlphaFormer's generated factors, we conduct backtests using a "top-$k$/drop-$n$" investment strategy during the testing period (2022-01-01, 2022-12-31) on the CSI300 dataset. On each trading day, we rank stocks by their alpha values and select the top $k$ stocks for an equally weighted portfolio. To minimize excessive trading costs, we limit the strategy to buying or selling at most $n$ stocks per day. Consistent with prior studies Yu et al. (2023); Shi et al. (2025), we set $k = 50$ and $n = 5$. The transaction cost is set to be $0.05\%$.

Figure 4 illustrates that the portfolio strategy based on AlphaFormer's generated factors outperforms the baseline methods on the CSI300 benchmark, which demonstrates its superior profitable ability.

## 5 Conclusion

In this paper, we propose AlphaFormer, a Transformer model designed to generate alpha factors from raw stock market data end-to-end. We also introduce a novel generative framework that integrates multiple time-series generative models, ensuring the creation of high-fidelity synthetic datasets essential for pre-training AlphaFormer. Extensive experiments demonstrate that AlphaFormer outperforms existing methods on real-world stock market datasets with lower inference costs. Simulated trading experiments also show its potential in practical applications.

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

## A   RELATED WORK

**Machine Learning-based Alpha Factors.** Machine learning-based methods for alpha mining have rapidly evolved, initially treating stock trends as individual time series predicted by models such as Multilayer Perceptrons (MLPs) LeCun et al. (2015), Transformers Vaswani et al. (2017), LSTMs Hochreiter & Schmidhuber (1997), and tree-based methods like LightGBM Ke et al. (2017) and XGBoost Chen & Guestrin (2016). Subsequent advancements introduced specialized architectures, exemplified by the SFM model Zhang et al. (2017) which employed a Discrete Fourier Transform (DFT)-like mechanism to identify multi-frequency trading patterns. More recently, the field has expanded to incorporate non-standard data sources for enhanced predictive power; for instance, Zolfagharinia et al. (2024) fuse multiple textual features from tweets to enhance stock price prediction, while HIST Xu et al. (2021) integrated concept graphs with time series to model shared characteristics and semantic relationships among stock groups. While these machine learning-based methods offer strong predictive capabilities, they often lack the interpretability of Formulaic Alpha methods.

**Formulaic Alpha Factors.** The search space for formulaic alpha factors is vast, given the extensive range of operators and features available. Traditionally, genetic programming has been used to generate these factors by mutating expression trees—structures representing mathematical expressions Koza (1994). For instance, Lin et al. (2019b) enhanced the gplearn library with time-series operators tailored to formulaic alphas, laying the foundation for an alpha-mining framework. This framework was advanced by Lin et al. (2019a), who enabled the mining of alphas with nonlinear relationships to returns, using the mutual Information Coefficient (IC) as the fitness metric. To improve the diversity of generated alpha sets, Zhang et al. (2020) applied mutual IC to filter out overly similar alphas. AlphaEvolve Cui et al. (2021) utilized computation graphs to represent alphas, enabling more complex operations like matrix-wise computations. It evolved new factors by building on existing ones through evolutionary algorithms. Recently, reinforcement learning emerged as a powerful approach to alpha generation. AlphaGen Yu et al. (2023) employed a policy network, implemented with recurrent neural networks (RNNs), to produce alpha expressions. Trained via reinforcement learning, the network received rewards based on the Information Coefficient (IC) of the generated alphas, encouraging the creation of highly predictive factors. Similarly, AlphaForge Shi et al. (2025) used a generative predictive neural network to generate factors, guided by a separate predictive network that acted as a reward model. This design helped the system learn effective factor generation strategies even with sparse reward signals. However, a key limitation persists in these methods: they start from scratch for each new alpha mining task, failing to leverage knowledge from prior efforts.

**Pre-trained Symbolic Regression.** Pre-trained Symbolic Regression (SR) commonly involves pre-training an encoder-decoder Transformer model for the end-to-end generation of symbolic expressions from input data. By pre-training on large-scale synthetic datasets, these Transformer models can generalize to unseen datasets and perform efficient inference, as task-specific retraining is often minimized or unnecessary. The work by Biggio et al. (2021); Kamienny et al. (2022); Valipour et al. (2021) demonstrated the effectiveness of pre-trained SR, achieving results comparable to GP methods while requiring significantly less inference computation. Holt et al. (2023) proposed an end-to-end loss function inspired by Bayesian inference, which helps the model learn equation invariances more effectively than traditional cross-entropy loss. With its demonstrated capabilities, pre-trained SR has found applications in diverse scientific areas, including the discovery of ordinary differential equations d'Ascoli et al. (2023a); Becker et al. (2023), Boolean formulas d'Ascoli et al. (2023b), and recurrence relations d'Ascoli et al. (2022). While not yet applied to alpha factor mining, the principles of pre-trained SR offer a promising direction for addressing the limitations of existing alpha generation methods, particularly in leveraging prior knowledge and improving efficiency.

## B   LIST OF OPERATORS, FEATURES, AND CONSTANTS

**Operators.** Our model utilizes four types of operators, categorized into two main groups: elementary operators and time-series operators. Elementary operators (denoted "E") process data exclusively from the current trading day. In contrast, time-series operators (denoted "TS") consider data over a consecutive period. Each group is further divided into unary operators (denoted "U"), which apply to a single series, and binary operators (denoted "B"), which apply to two series.

Table 4: Table of Operators

| Operator | Category | Description |
|---|---|---|
| $\text{Abs}(x)$ | E-U | The absolute value, $|x|$. |
| $\text{Log}(x)$ | E-U | The natural logarithm, $\ln(x)$. |
| $x+y, x-y, x \cdot y, x/y$ | E-B | Standard arithmetic operations: addition, subtraction, multiplication, and division. |
| $\text{Greater}(x,y), \text{Less}(x,y)$ | E-B | Returns the greater or lesser of the two values $x$ and $y$, respectively. |
| $\text{Ref}(x,t)$ | TS-U | The value of expression $x$ evaluated $t$ days prior to the current day. |
| $\text{Mean}(x,t), \text{Med}(x,t), \text{Sum}(x,t)$ | TS-U | The arithmetic mean, median, or sum of expression $x$ over the preceding $t$ days. |
| $\text{Std}(x,t), \text{Var}(x,t)$ | TS-U | The sample standard deviation or variance of expression $x$ over the preceding $t$ days. |
| $\text{Max}(x,t), \text{Min}(x,t)$ | TS-U | The maximum or minimum value of expression $x$ over the preceding $t$ days. |
| $\text{Mad}(x,t)$ | TS-U | The Mean Absolute Deviation of $x$ over the preceding $t$ days, defined as $\mathbb{E}[|x_i - \mathbb{E}[x]|]$. |
| $\text{Delta}(x,t)$ | TS-U | The difference $x_{\text{current}} - \text{Ref}(x,t)$. |
| $\text{WMA}(x,t), \text{EMA}(x,t)$ | TS-U | Weighted Moving Average (WMA) or Exponential Moving Average (EMA) of $x$ over the preceding $t$ days. |
| $\text{Cov}(x,y,t)$ | TS-B | The sample covariance between time series $x$ and $y$ over the preceding $t$ days. |
| $\text{Corr}(x,y,t)$ | TS-B | The sample Pearson correlation coefficient between time series $x$ and $y$ over the preceding $t$ days. |

Table 5: Table of Stock Features

| Feature | Description |
|---|---|
| Open | The opening price of the stock for the trading day. |
| Close | The closing price of the stock for the trading day. |
| High | The highest price of the stock during the trading day. |
| Low | The lowest price of the stock during the trading day. |
| Volume | The total number of shares traded during the trading day. |
| VWAP | The volume-weighted average price of the stock during the trading day. |

Table 6: Table of Constants

| Category | Values |
|---|---|
| Basic | $-30, -10, -5, -2, -1, -0.5, -0.01, 0.01, 0.5, 1, 2, 5, 10, 30$ |
| Time-Series | $10d, 20d, 30d, 40d, 50d$ |

**Features.** The model utilizes six features recorded daily for each stock. These features are described in Table 5.

**Constants.** Our model employs two categories of constants. 'Basic' constants are utilized within elementary operators, while 'Time-Series' constants specify the lookback period (time range) for time-series operators. The permissible values for these constants are detailed in Table 6.

## C    DETAILS OF GENERATING SYNTHETIC STOCK DATASETS

**GRU.** GRU Dey & Salem (2017) is a variant of recurrent neural networks (RNNs) designed to model sequential data by maintaining a hidden state that encapsulates temporal dependencies. It utilizes two key gating mechanisms—the update gate and the reset gate—to regulate the flow of information, enabling the model to selectively retain relevant past information while integrating new observations. In time-series generation, the GRU leverages its hidden state, which encodes the sequence history up to the current time step, to condition on past data. The generation process operates autoregressively: the model predicts the next time step based on the current hidden state, feeds this prediction back as input, and iterates to produce subsequent future values, thereby generating a sequence of future data conditioned on the observed past.

**Transformer.** Transformer Vaswani et al. (2017) is a neural network architecture that employs a attention mechanism to process sequential data, allowing it to weigh the relevance of all parts of the input sequence simultaneously and capture long-range dependencies effectively. Unlike recurrent models, it processes the entire sequence in parallel, making it well-suited for time series with intricate, extended patterns. For time-series generation, the Transformer conditions on past data through masked attention, which restricts the model to attend only to previous time steps when predicting future values. During generation, it operates autoregressively by predicting the next time step based on the historical sequence, incorporating each prediction into the input for the subsequent step, and repeating this process to construct a future sequence that reflects the context of the past data.

**CSDI.** CSDI (Conditional Score-based Diffusion Model) Tashiro et al. (2021) is a generative model that employs diffusion processes to model time-series data distributions probabilistically. Diffusion models function by incrementally adding noise to data and learning to reverse this process, thereby reconstructing the original distribution. In time-series generation, CSDI conditions on past data by integrating observed values into the diffusion process, enabling the model to generate future sequences consistent with the historical context. The generation involves sampling from the conditional distribution of future time steps given the observed past, guided by a score-based approach. Through iterative denoising steps, CSDI refines an initial noisy sequence into coherent future data.

**LSTM Evaluator.** The LSTM Evaluator is a Long Short-Term Memory network, a specialized type of recurrent neural network designed to model sequential data by capturing long-term temporal dependencies. It employs memory cells and three key gating mechanisms—the input gate, forget gate, and output gate—to regulate the retention and flow of information over time. Trained on real-world stock data, the LSTM Evaluator learns to model the conditional probability distribution $P(x_t|x_1,\ldots,x_{t-1})$ for each time step $t$. For a generated future sequence $x_{c+1}, x_{c+2}, \ldots, x_{c+T}$ conditioned on a historical context $x_1, x_2, \ldots, x_c$, the LSTM Evaluator computes the likelihood as the product of conditional probabilities: $\prod_{t=1}^{T} P(x_{c+t}|x_1,\ldots,x_{c+t-1})$. This likelihood score determines the plausibility of the synthetic sequence, guiding the selection of the highest-quality data for the synthetic dataset.

All models, including the generative models and the LSTM Evaluator, were trained on real-world stock data from the Chinese stock market, specifically stocks listed in the CSI 300 and CSI 500 indices, spanning 2012-01-01 to 2021-12-31, totaling 1,868 unique stocks. The generative models were trained to produce future stock data conditioned on historical sequences, while the LSTM Evaluator was trained to model conditional distribution, enabling it to compute likelihoods for evaluating generated sequences. After training, each generative model was conditioned on historical segments of 200 time steps to generate synthetic future sequences, following the procedures outlined in subsection 3.1. Through this approach, 1,000 synthetic stock datasets (containing approximately $550K$ individual synthetic stock sequences) were generated for pre-training. The pseudo-code for synthetic dataset generation is provided in Algorithm 2, with the corresponding hyperparameters listed in Table 7.

## D    DETAILS OF MODEL ARCHITECTURE

Our model has three major components: Dataset Embedder for dataset embedding, LSTM Encoder for factor embeddings, and encoder-decoder Transformer for factor generation. The hyperparameters configuration can be found in Table 8.

Table 7: Synthetic Dataset Generation Hyperparameters

| Hyperparameter | Value |
|---|---|
| $S_{\min}$ | 300 |
| $S_{\max}$ | 800 |
| $T_{\min}$ | 500 |
| $T_{\max}$ | 1000 |
| $c$ | 200 |

---

**Algorithm 2** Process of Generating Synthetic Datasets

---

1: **Input:** Generative models $\{G_i\}_{i=1}^{k}$, Evaluator $E$, Real-world stock data $\mathcal{R}$, Hyperparameters $S_{\min}, S_{\max}, T_{\min}, T_{\max}, c$
2: **Output:** A synthetic stock dataset $\mathcal{D}$
3: Sample $S \sim \mathrm{Uniform}[S_{\min}, S_{\max}]$
4: Sample $T \sim \mathrm{Uniform}[T_{\min}, T_{\max}]$
5: $\mathcal{D} \leftarrow \emptyset$
6: **for** $s = 1$ to $S$ **do**
7:     Sample context $\mathrm{context}_s$ of size $c$ from $\mathcal{R}$
8:     Generate $\{\mathrm{sequence}_i = G_i(\mathrm{context}_s) \mid i = 1, \ldots, k\}$
9:     Compute $l_i = E(\mathrm{sequence}_i \mid \mathrm{context}_s)$ for $i = 1, \ldots, k$
10:    $j \leftarrow \arg\max_{i=1,\ldots,k} l_i$
11:    $\mathrm{sequence}_s \leftarrow \mathrm{sequence}_j$
12:    $\mathcal{D} \leftarrow \mathcal{D} \cup \{\mathrm{sequence}_s\}$
13: **return** $\mathcal{D}$

---

Table 8: Model Architecture Hyperparameters

| Component | Module / Hyperparameter | Value |
|---|---|---|
| **Dataset Embedder** | LSTM: Number of Layers | 2 |
| | LSTM: Hidden Size ($n_{hid}$) | 128 |
| | Transformer Encoder: Number of Layers | 2 |
| | Transformer Encoder: Hidden Size | 128 |
| **LSTM Encoder** | Number of Layers | 2 |
| | Hidden Size ($n_{hid}$) | 128 |
| **Encoder-Decoder Transformer** | Encoder Layers | 2 |
| | Decoder Layers | 4 |
| | FFN Dimension | 512 |

## E  DETAILS OF PRE-TRAINING

Our objective is to pre-train a conditional alpha factor generator $P_\theta(f|\mathcal{D}, \mathcal{P})$ that generates new alpha factors $f$ to iteratively refine an Alpha Pool $\mathcal{P}$ for a given dataset $\mathcal{D}$. The pre-training procedure is structured as follows:

1. **Initialization**: For each dataset $\mathcal{D}_i$ in a batch of $k$ datasets, we randomly sample a maximum Alpha Pool size $M$ from the set $\{1, 10, 20\}$. This variability promotes generalization across different pool capacities. We then initialize an empty Alpha Pool $\mathcal{P}_0^i$ with maximum size $M$ and set the number of iterations, num_iter, to $2M$, ensuring sufficient iterations to learn both the addition and pruning of alpha factors.

2. **Experience Collection**: For each iteration $j$ from 1 to num_iter:
   - Sample a new alpha factor $f_j \sim P_\theta(f|\mathcal{D}_i, \mathcal{P}_{j-1}^i)$ using the current generator.
   - Update the Alpha Pool to $\mathcal{P}_j^i$ by applying a specified update mechanism (e.g., adding $f_j$ and potentially pruning less effective factors).

Table 9: Pre-training Hyperparameters

| Hyperparameter | Value |
|---|---|
| Learning Rate | 0.0003 |
| L1 Coefficient | 0.005 |
| PPO Clip Ratio | 0.2 |
| Value Function Coefficient | 0.5 |
| Max Gradient Norm | 0.5 |
| Batch Size | 100 |
| Epoch | 10 |

- Compute the reward $r_j$ as the Information Coefficient (IC) of the updated pool $\mathcal{P}_j^i$ on $\mathcal{D}_i$.
- Store the experience tuple $e_j = (\mathcal{D}_i, \mathcal{P}_{j-1}^i, f_j, r_j)$ in a replay buffer $\mathcal{B}$.

3. **Model Optimization**: Using the Proximal Policy Optimization (PPO) algorithm, we update the model parameters after collecting experiences. For each mini-batch sampled from $\mathcal{B}$, we compute the total loss as follows:

$$\mathcal{L}(\theta, \phi) = \mathcal{L}^{\text{CLIP}}(\theta) + \eta \mathcal{L}^{\text{value}}(\phi) \tag{5}$$

where $\mathcal{L}^{\text{CLIP}}(\theta)$ is the clipped policy loss:

$$\mathcal{L}^{\text{CLIP}}(\theta) = -\hat{\mathbb{E}}\left[\min\left\{\text{ratio}(\theta)\hat{A}, \text{clip}\left(\text{ratio}(\theta), 1-\epsilon, 1+\epsilon\right)\hat{A}\right\}\right], \tag{6}$$

where $\hat{A}$ is calculated by $r - V_\phi(\mathcal{D}, \mathcal{P})^2$. And $\mathcal{L}^{\text{value}}(\phi)$ is the value function loss:

$$\mathcal{L}^{\text{value}}(\phi) = \|V_\phi(\mathcal{D}, \mathcal{P}) - r\|_2^2. \tag{7}$$

We minimize the total loss using gradient descent with respect to both $\theta$ and $\phi$, enhancing the generator's ability to produce effective alpha factors.

A comprehensive list of hyperparameters for the pre-training phase is provided in Table 9. The pseudo-code of pre-training can be found in Algorithm 3.

## F  EXAMPLE OF ALPHA POOL

| # | Alpha | Weight |
|---|---|---|
| 1 | Div(Div(1.0,Div(Sub(high,Abs(Max(Mul(30.0,5.0),20d)))),0.01)),-0.5) | -0.0043 |
| 2 | Delta(Log(close),40d) | -0.0148 |
| 3 | Less(Abs(Div(-10.0,Abs(-10.0))),Abs(Var(volume,20d))) | -0.0054 |
| 4 | Less(Add(Abs(close),Mul(low,Sub(-2.0,Add(-1.0,-0.5)))),Abs(30.0)) | -0.0102 |
| 5 | Mul(Mul(Std(Log(close),5d),Med(-10.0,10d)),-0.5) | 0.0102 |
| 6 | Log(Greater(Log(volume),-10.0)) | -0.0237 |
| 7 | Std(Greater(Div(Abs(-0.5),0.01),open),10d) | -0.0059 |
| 8 | Less(Std(Corr(low,high,5d),20d),30.0) | -0.0080 |
| 9 | EMA(Mul(0.5,Abs(Mul(volume,vwap))),20d) | -0.0085 |
| 10 | Abs(Mul(low,Sub(low,Abs(-30.0)))) | -0.0035 |
| 11 | Min(Div(close,Greater(Add(-5.0,Mul(Corr(0.5,volume,10d),10.0)),low)),5d) | -0.0058 |
| 12 | Div(Var(Log(volume),10d),30.0) | -0.0091 |
| 13 | Mean(Ref(Abs(Min(Sub(close,low),20d)),10d),10d) | 0.0050 |
| 14 | Less(low,Mul(Log(Div(high,Add(low,2.0))),Abs(1.0))) | -0.0566 |
| 15 | Delta(Log(high),20d) | -0.0430 |

*Continued on next page*

---

[2]The value function $V_\phi(\mathcal{D}, \mathcal{P})$ is computed by a two-layer MLP value head, which takes encoded representations from the generator's transformer encoder as input. As a result, some parameters of the value function are shared with the generator.

| # | Alpha | Weight |
|---|---|---|
| | *Continued from previous page* | |
| 16 | Abs(Abs(Std(WMA(Greater(5.0,Log(volume)),10d),10d))) | -0.0034 |
| 17 | Add(Add(Sub(Less(Sub(low,Div(0.01,-2.0)),5.0),-10.0),close),-0.5) | -0.0215 |
| 18 | Mul(Div(Ref(Sum(WMA(Greater(30.0,close),10d),5d),10d),-5.0),open) | 0.0094 |
| 19 | Cov(close,Var(Add(Log(5.0),Greater(volume,Greater(5.0,open))),20d),10d) | 0.0070 |
| 20 | Corr(Add(close,-2.0),vwap,20d) | 0.0216 |
| 21 | Corr(Abs(Greater(Abs(Add(0.01,Log(Min(volume,1d))))),-5.0)),volume,40d) | 0.0083 |
| 22 | Var(Log(Mad(Ref(Greater(volume,-30.0),5d),5d)),10d) | -0.0083 |
| 23 | Delta(Abs(Mean(Mul(Greater(Less(10.0,Abs(open)),-1.0),2.0),5d)),5d) | 0.0033 |
| 24 | Mean(Abs(Corr(vwap,close,10d)),5d) | 0.0117 |
| 25 | Delta(low,20d) | -0.0042 |
| 26 | Log(Std(Mul(Add(-5.0,Greater(2.0,-5.0)),vwap),5d)) | 0.0094 |
| 27 | Mul(Abs(Max(Delta(Abs(Corr(low,vwap,40d)),1d),1d)),-1.0) | 0.0031 |
| 28 | Abs(Med(Max(Div(Mul(0.01,-0.5),Greater(volume,Add(30.0,-0.01))),40d),1d)) | 0.0177 |
| 29 | Std(Div(Greater(Var(-0.5,5d),close),open),10d) | 0.0108 |
| 30 | Mul(Cov(Div(low,Greater(low,2.0)),close,20d),Min(Greater(0.5,0.01),10d)) | 0.0029 |
| 31 | Var(Greater(Less(Div(volume,-10.0),Div(0.01,30.0)),-30.0),5d) | -0.0044 |
| 32 | Corr(low,Sub(2.0,close),10d) | -0.0093 |
| 33 | Less(Div(Min(Abs(Div(-30.0,Add(Less(0.01,-30.0),close))),1d),1.0),low) | -0.0027 |
| 34 | WMA(Delta(Sum(volume,10d),10d),10d) | -0.0041 |
| 35 | Div(Ref(Sub(Greater(Med(Greater(close,vwap),1d),-0.5),1.0),5d),open) | 0.0315 |
| 36 | Mean(Corr(vwap,volume,20d),10d) | -0.0231 |
| 37 | Greater(Delta(vwap,1d),-30.0) | -0.0045 |
| 38 | Sub(2.0,Sub(-1.0,Delta(Corr(high,close,10d),20d))) | -0.0073 |
| 39 | Abs(Sub(Abs(Abs(Sub(Abs(-2.0),Add(0.5,Abs(high))))),low)) | -0.0099 |
| 40 | Less(vwap,Greater(Log(Delta(vwap,20d)),-1.0)) | -0.0110 |
| 41 | Less(Delta(vwap,40d),5.0) | 0.0182 |
| 42 | Delta(Abs(Div(Abs(30.0),Greater(-10.0,Log(close)))),40d) | -0.0067 |
| 43 | Abs(Cov(Mul(Std(high,20d),-10.0),Max(Sum(high,1d),40d),5d)) | 0.0041 |
| 44 | Less(Ref(Mul(Add(volume,-0.01),10.0),5d),Corr(-0.5,-2.0,40d)) | 0.0067 |
| 45 | Abs(Log(Mean(Add(-2.0,volume),40d))) | -0.0155 |
| 46 | Sum(Div(Div(Mul(Mul(1.0,Div(open,0.01)),-10.0),Abs(0.01)),vwap),5d) | 0.0145 |
| 47 | Log(Abs(Sub(Div(Min(Div(WMA(volume,10d),-0.01),40d),-2.0),Log(0.01)))) | -0.0051 |
| 48 | Div(EMA(open,40d),Abs(vwap)) | -0.0302 |
| 49 | Cov(Add(Min(Mul(1.0,WMA(close,5d)),1d),-2.0),volume,10d) | -0.0033 |
| 50 | Mul(Log(Sub(Greater(Greater(Min(volume,10d),0.5),1.0),Mul(vwap,30.0))),30.0) | -0.0086 |
| 51 | Mad(Log(Abs(Sub(30.0,close))),40d) | -0.0096 |
| 52 | Var(Log(Abs(high)),5d) | -0.0034 |
| 53 | Less(Log(volume),Div(vwap,low)) | 0.0070 |
| 54 | Abs(Mul(Sub(Abs(Greater(Greater(open,-0.5),-1.0)),close),-2.0)) | 0.0036 |
| 55 | WMA(Corr(close,Mul(Add(Div(Add(0.5,low),-30.0),-2.0),-2.0),40d),5d) | 0.0078 |
| 56 | Div(low,Mad(Sub(close,0.5),40d)) | 0.0047 |
| 57 | Greater(Log(2.0),Corr(volume,high,20d)) | -0.0146 |
| 58 | Div(Greater(Log(Log(high)),-30.0),0.5) | -0.0030 |
| 59 | Less(Delta(close,20d),WMA(1.0,10d)) | 0.0046 |
| 60 | Add(Add(Log(open),-5.0),Mul(close,-0.01)) | -0.0466 |
| 61 | Mul(Div(Add(Delta(Greater(-0.01,Abs(vwap)),5d),2.0),open),high) | -0.0075 |
| 62 | Less(Log(Abs(volume)),Mean(Sub(Mul(close,vwap),-5.0),1d)) | -0.0114 |
| 63 | Sub(Greater(-5.0,Abs(Log(vwap))),Log(high)) | 0.0105 |
| 64 | Add(Add(Corr(vwap,Mean(low,1d),5d),Add(Mean(-0.5,20d),-5.0)),-0.01) | 0.0097 |
| 65 | Cov(Log(high),Mul(volume,Add(10.0,-1.0)),20d) | -0.0047 |
| 66 | Mul(Add(Div(Ref(open,1d),high),0.01),5.0) | 0.0133 |
| 67 | Var(Add(Abs(Delta(open,1d)),-0.5),5d) | -0.0061 |
| 68 | Corr(Div(open,-30.0),volume,5d) | -0.0036 |
| | *Continued on next page* | |

| # | Alpha | Weight |
|---|---|---|
| | *Continued from previous page* | |
| 69 | Less(Ref(volume,20d),10.0) | 0.0043 |
| 70 | Corr(Mul(-2.0,Abs(low)),Div(volume,vwap),5d) | 0.0130 |
| 71 | Div(low,WMA(low,20d)) | -0.0218 |
| 72 | Greater(Greater(0.5,Corr(low,Sub(-2.0,volume),40d)),-0.5) | -0.0044 |
| 73 | Greater(Mul(Ref(Mad(close,40d),1d),5.0),-5.0) | 0.0097 |
| 74 | Mul(EMA(Sub(Delta(high,1d),1.0),10d),Mul(2.0,close)) | 0.0027 |
| 75 | Std(Div(Less(close,10.0),Sub(1.0,2.0)),20d) | 0.0131 |
| 76 | Abs(Delta(Abs(volume),40d)) | -0.0077 |
| 77 | Abs(Add(Div(vwap,low),Less(Add(2.0,0.01),-10.0))) | -0.0070 |
| 78 | Less(Div(Min(Abs(Greater(0.01,Div(low,0.5))),20d),Less(open,30.0)),high) | -0.0186 |
| 79 | Delta(Div(-5.0,high),20d) | 0.0060 |
| 80 | Delta(Sub(Abs(Abs(-1.0)),Sum(open,20d)),20d) | 0.0137 |
| 81 | Abs(Div(Delta(Greater(Abs(Div(-30.0,low)),-2.0),1d),2.0)) | 0.0043 |
| 82 | Abs(Delta(WMA(Std(Add(vwap,Log(30.0)),5d),10d),20d)) | -0.0046 |
| 83 | Mul(Max(Sub(vwap,high),10d),Mul(Log(Sum(30.0,5d)),2.0)) | 0.0105 |
| 84 | Div(Mul(WMA(close,5d),Mul(-0.5,30.0)),vwap) | 0.0289 |
| 85 | Add(Abs(Log(Add(Abs(Sub(0.01,volume)),Mul(-1.0,-0.5)))),open) | 0.0910 |
| 86 | Ref(Add(Sub(Mad(Greater(Log(Log(low)),30.0),5d),10.0),volume),40d) | 0.0030 |
| 87 | Abs(Delta(Mul(Sub(Mul(Delta(-5.0,1d),0.01),close),0.5),40d)) | 0.0210 |
| 88 | Corr(close,vwap,10d) | 0.0044 |
| 89 | Delta(Greater(vwap,30.0),20d) | 0.0110 |
| 90 | Div(Log(Mul(Std(Div(vwap,close),10d),5.0)),-10.0) | 0.0072 |
| 91 | Min(Add(Cov(low,close,5d),close),1d) | -0.0111 |
| 92 | Less(Div(WMA(Min(open,10d),20d),vwap),10.0) | -0.0184 |
| 93 | Div(Log(Greater(Mul(EMA(-0.01,10d),-5.0),Sub(5.0,Abs(low)))),-0.01) | -0.0088 |
| 94 | Mul(Abs(Log(Greater(-1.0,Greater(Abs(volume),1.0)))),30.0) | 0.0060 |
| 95 | Cov(Add(Abs(Abs(close)),30.0),volume,20d) | -0.0030 |
| 96 | Div(vwap,Std(Log(Div(0.5,volume)),20d)) | -0.0130 |
| 97 | Div(vwap,Sub(Mul(close,Less(vwap,-10.0)),1.0)) | -0.0220 |
| 98 | Corr(close,volume,5d) | -0.0072 |
| 99 | Delta(Mul(WMA(volume,10d),Greater(Less(-5.0,Min(volume,10d)),low)),5d) | -0.0035 |
| 100 | Less(Sum(volume,20d),Abs(Mul(-0.01,Less(-5.0,0.5)))) | -0.0059 |

Table 10 presents a representative combination of 100 alpha factors generated by our framework and evaluated on the CSI300 constituent stocks. The weights are computed by minimizing Eq 2. These factors are obtained through an iterative refinement process applied to the $20,000$ alpha factors produced by AlphaFormer, capturing the distilled essence of the generated factors.

## G  BASELINES

In our experiments, we evaluate our approach against baselines from two distinct paradigms: machine learning-based methods and symbolic regression-based methods. Machine learning-based methods directly predict stock trends, whereas symbolic regression-based methods generate alpha factors for use in predictive models. Detailed descriptions of each baseline are provided below.

1. **Machine Learning-based Methods**

    **XGBoost:** A gradient boosting framework that leverages an ensemble of decision trees to predict stock trends efficiently. Implemented using the xgboost Python library[3].

    **LightGBM:** A gradient boosting framework optimized for speed and scalability, utilizing tree-based algorithms to predict stock trends. Implemented using the lightgbm Python library[4].

---

[3] https://xgboost.readthedocs.io
[4] https://lightgbm.readthedocs.io

Table 11: Hyperparameter settings for Symbolic Regression and Machine Learning-based methods.

**Symbolic Regression Methods**

**GP Hyperparameters**

| Parameter | Value |
|---|---|
| population_size | 1000 |
| generations | 40 |
| init_depth | (2, 6) |
| tournament_size | 600 |
| p_crossover | 0.3 |
| p_subtree_mutation | 0.1 |
| p_hoist_mutation | 0.01 |
| p_point_mutation | 0.1 |
| p_point_replace | 0.6 |
| max_samples | 0.9 |
| L1 Coefficient | 0.005 |

**AlphaGen Hyperparameters**

| Parameter | Value |
|---|---|
| LSTM Num Layers | 2 |
| LSTM Hidden Dimension | 128 |
| Value Head Num Layers | 2 |
| Value Head Dimension | 64 |
| Policy Head Num Layers | 2 |
| Policy Head Dimension | 64 |
| Clip Range | 0.2 |
| Learning Rate | 0.001 |
| L1 Coefficient | 0.005 |

**AlphaForge Hyperparameters**

| Parameter | Value |
|---|---|
| Generator Num Layers | 2 |
| Generator Hidden Dimension | 128 |
| Predictor Num Layers | 2 |
| Predictor Hidden Dimension | 128 |
| Correlation Threshold | 0.8 |
| IC Threshold | 0.025 |
| ICIR Threshold | 0.1 |
| Learning Rate | 0.001 |
| L1 Coefficient | 0.005 |

**Machine Learning-based Methods**

**LightGBM Hyperparameters**

| Parameter | Value |
|---|---|
| Num Leaves | 210 |
| Max Depth | 8 |
| Learning Rate | 0.05 |
| Feature Fraction | 0.9 |
| Bagging Fraction | 0.8 |
| Bagging Frequency | 5 |

**XGBoost Hyperparameters**

| Parameter | Value |
|---|---|
| Colsample_bytree | 0.9 |
| Max Depth | 8 |
| Learning Rate | 0.05 |
| Subsample | 0.9 |
| Num Boost Rounds | 1000 |

**MLP Hyperparameters**

| Parameter | Value |
|---|---|
| Num Layers | 2 |
| Batch Size | 512 |
| Learning Rate | 0.001 |
| Hidden Dimension | 64 |

**LSTM Hyperparameters**

| Parameter | Value |
|---|---|
| Num Layers | 2 |
| Batch Size | 512 |
| Learning Rate | 0.001 |
| Hidden Dimension | 64 |

---

**Algorithm 3** Pre-training Procedure for Conditional Alpha Factor Generator

---

1: **Input:** Batch size $k$ of datasets, Conditional generator $P_\theta(f|\mathcal{D}, \mathcal{P})$ , Value function $V_\phi(\mathcal{D}, \mathcal{P})$ , Loss function $\mathcal{L}(\theta, \phi)$ defined by Equation (5) for pre-training , Hyperparameters (e.g., learning rate $lr$)
2: **Output:** Pre-trained conditional generator $P_\theta(f|\mathcal{D}, \mathcal{P})$
3: **Initialization:**
4: Initialize replay buffer $\mathcal{B}$
5: **Experience Collection:**
6: **for** $i$ in $\{1, ..., k\}$ **do**
7:     $M \leftarrow$ sample from $\{1, 10, 20\}$
8:     Initialize $\mathcal{P}_0^i$ as empty Alpha Pool with maximum size $M$
9:     num_iter $\leftarrow 2 \times M$
10:     **for** $j$ in $\{1, ..., \text{num\_iter}\}$ **do**
11:         $f_j \sim P_\theta(f|\mathcal{D}_i, \mathcal{P}_{j-1}^i)$            ▷ Sample alpha factor from conditional generator
12:         $\mathcal{P}_j^i \leftarrow \text{update}(\mathcal{P}_{j-1}^i, f_j)$        ▷ Update the Alpha Pool based on Algorithm 1
13:         $r_j \leftarrow \mathcal{R}(\mathcal{P}_j^i, \mathcal{D}_i)$             ▷ Calculate reward based on Equation (3)
14:         $e_j \leftarrow (\mathcal{D}_i, \mathcal{P}_{j-1}^i, f_j, r_j)$         ▷ Collect experience tuple
15:         $\mathcal{B} \leftarrow \mathcal{B} \cup e_j$
16: **Model Optimization:**
17: **for** each mini-batch $B$ from $\mathcal{B}$ **do**
18:     $L \leftarrow 0$
19:     **for** each experience tuple $e$ from $B$ **do**
20:         $L \leftarrow L + \mathcal{L}(\theta, \phi)$        ▷ Calculate loss function based on experience tuple
21:     $\theta \leftarrow \theta - lr \times \nabla_\theta L$
22:     $\phi \leftarrow \phi - lr \times \nabla_\phi L$

---

    **MLP:** A multi-layer perceptron, a feedforward neural network with non-linear activation functions, designed to model complex relationships in input features for stock trend prediction. Implemented using the pytorch Python library[5].

2. **Symbolic Regression-based Methods**

    **GP:** Genetic Programming, an evolutionary algorithm that derives mathematical expressions to serve as alpha factors. Implemented using the gplearn Python library[6].

    **AlphaGen:** A framework employing reinforcement learning to identify a synergistic set of alpha factors. Implemented using the authors' open-source code[7].

    **AlphaForge:** A framework that uses a generative predictive neural network to produce alpha factors. Implemented using the authors' open-source code[8].

# H   Hardware Specification

Our experiments were performed on a Linux-based system equipped with an AMD EPYC 9654 96-Core Processor and an NVIDIA L20 GPU.

# I   Limitation

Our approach for alpha mining relies on GPU for training and inference, while traditional methods like GP operate in CPU environments. Consequently, our method may not be well-suited for deployment in CPU environments.

---

[5] https://pytorch.org/
[6] https://gplearn.readthedocs.io
[7] https://github.com/RL-MLDM/alphagen
[8] https://github.com/dulyhao/alphaforge

## J  BROADER IMPACT

**Social Impact.** This paper investigates alpha mining, leveraging AlphaFormer for end-to-end generation of alpha factors. This approach enables the efficient discovery of novel, exploitable alpha factors, enhancing investment strategies and potentially democratizing access to advanced financial tools.

