# OpenReview forum: "AlphaFormer: End-to-End Symbolic Regression of Alpha Factors with Transformers"
_ICLR.cc/2026/Conference — ICLR 2026 Conference Withdrawn Submission_

### Official Review · Reviewer_kFA2 · 2025-10-30

**Soundness:** 2
**Presentation:** 3
**Contribution:** 2
**Rating:** 4
**Confidence:** 4

**Summary:**

This paper proposes AlphaFormer to learn alpha factors from raw stock market data, lying in the symbolic regression scheme. It also incorporates with a pre-training process on synthetic data to transfer prior knowledge. AlphaFormer is claimed to outperform alternative baselines in terms of both evaluation metrics and simulated trading performance.

**Strengths:**

- The use of pre-training to transfer knowledge for symbolic alpha discovery is interesting.

- The preliminaries about alpha factor, alpha mining, factor pools, and evaluation are clearly formalized, making the paper more readable to a broader ML audience.

**Weaknesses:**

- The synthetic data generation may not provide sufficient information to learn prior knowledge for transformation into downstream applications. The generative models used are not specifically designed for financial data, which probably fail to capture useful temporal dynamics but output some noise. Moreover, the likelihood-based selection of high-quality synthetic data cannot guarantee diversity.

- This paper claims that the proposed approach reduces inference computation, but does not provide its complexity (or runtime) and the comparison against existing methods.

- [Minor] Citation formatting appears incorrect throughout the paper.

**Questions:**

- Beyond CSDI used in this paper, there are some other representative deep generative models particularly designed for time series, such as TimeGAN [1] and Diffusion-TS [2]. Have they been explored for the synthetic data generation in this paper?

- As far as I know, vanilla GRU and Transformer without specific designs are not the common choice for effective time series generation. Why are they chosen for the synthetic data generation?

- What are the computational complexity and runtime of the proposed approach?

- Until 2021, the results in Figure 4 do not show that AlphaFormer outperforms the other baselines. Does it mean that the effectiveness of the proposed approach may be limited in some specific periods?



[1] Yoon et al. "Time-series generative adversarial networks." *Neurips* 2019.

[2] Yuan et al. “Diffusion-TS: Interpretable Diffusion for General Time Series Generation.“ *ICLR* 2024.

---

### Official Review · Reviewer_boST · 2025-10-31

**Soundness:** 2
**Presentation:** 2
**Contribution:** 2
**Rating:** 4
**Confidence:** 3

**Summary:**

The paper introduces AlphaFormer that performs symbolic regression to generate synergistic alpha factors. The model is pre-trained on synthetic time-series datasets produced by multiple generative models and then iteratively constructs an alpha pool on real data. Empirically, AlphaFormer achieves higher IC, Rank IC, Sharpe ratio, and annual return than several baselines, and delivers higher cumulative returns in backtests on CSI 300 and S&P 500.

**Strengths:**

- Deep learning for symbolic regression of alpha factors is well-motivated.

- AlphaFormer provides interpretable alpha factors while enabling efficient end-to-end generation.

- Heavy lifting in pre-training with a light, fixed-parameter inference loop is operationally attractive.

**Weaknesses:**

- The synthetic data generation part is somewhat unreliable. Alpha factors learned from synthetic data cannot generalize to the real stock market as long as the synthetic data do not fully resemble real data.

- The backtest curve in Figure 4 shows that until roughly 2021 the cumulative excess return of AlphaFormer is not strictly superior. The effectiveness of AlphaFormer in the simulated trading is weak.

**Questions:**

- How do you confirm that the prior knowledge learned from synthetic data is economically meaningful across regimes? Have you compared pre-training on synthetic data with pre-training on real data?

- Statistical methods, such as ARMA-GARCH and the bootstrap, are considered generative models in finance. Why not include these methods to generate synthetic data?

- The evaluation of synthetic data quality is model-centric. Have you considered statistical criteria?

- Why do you use an LSTM for stock embedding? It seems that transformers are more suitable for this task.

---

### Official Review · Reviewer_JWzm · 2025-11-01

**Soundness:** 3
**Presentation:** 3
**Contribution:** 2
**Rating:** 4
**Confidence:** 3

**Summary:**

This paper addresses a key limitation in alpha factor discovery: existing symbolic regression methods restart from scratch for each new dataset without leveraging prior knowledge. AlphaFormer proposes an encoder-decoder Transformer for end-to-end alpha factor generation from raw stock data, enabled by pre-training on synthetic datasets to learn transferable patterns. To generate high-fidelity synthetic stock data with temporal dependencies, the authors introduce a novel framework that integrates multiple time-series generative models (GRU, Transformer, Diffusion) with dynamic quality-based selection via an LSTM evaluator. The model architecture embeds both the dataset (using LSTM and Transformer encoders to capture cross-stock relationships) and the existing factor pool, then autoregressively generates new factors in Reverse Polish Notation. Experiments on CSI300 and CSI500 show AlphaFormer achieves superior IC (6.01% vs 5.19% for the best baseline) while generating only 33% as many factors and requiring no retraining during inference. Backtesting demonstrates the highest annual returns, highlighting practical potential through efficient pre-training and inference.

**Strengths:**

(1) Novel approach to synthetic data generation for financial time series
Traditional symbolic regression pre-training uses simple distributions (e.g., sampling independent points). However, AlphaFormer did:
(i) Temporal dependencies (ii) Cross-stock correlations: Dependencies between multiple stocks. The proposed multi-model generation framework with dynamic selection addresses this challenge effectively, representing a non-trivial contribution to generating realistic financial data for pre-training.

(2) Paradigm shift from search to conditional generation. This paper demonstrates substantial originality by reconceptualizing alpha factor discovery. While prior methods (GP-based and RL-based like AlphaGen) frame this as a search/optimization problem requiring extensive exploration per dataset, AlphaFormer transforms it into a conditional generation problem with pre-training. This enables knowledge transfer across datasets and significantly improves inference efficiency.

(3) Strong empirical results with practical implications. The method achieves state-of-the-art performance across multiple metrics (IC, Rank IC, Sharpe Ratio, CAGR) while generating 67% fewer factors than baselines. The cross-market generalisation and superior backtest returns demonstrate real-world applicability.

**Weaknesses:**

(1) Compared to AlphaGen, the innovation is not so significant. The main contributions are adding pre-training and multi-stock modelling. The primary improvements are replacing the RL training paradigm with pre-training and swapping LSTM for Transformer. While this brings significant efficiency gains, the core mechanisms remain similar to AlphaGen, including RPN representation, alpha pool management, and L1-regularised linear combination. The work feels more like an incremental extension ("AlphaGen + pre-training") rather than a fundamental methodological breakthrough.

(2) The paper claims "high-fidelity synthetic datasets," but provides insufficient validation: No distribution distance metrics (e.g., Wasserstein distance, MMD) between synthetic and real data are reported. No analysis of what temporal properties the generative models capture (autocorrelation, volatility clustering, etc.). Using likelihood for sample selection does not guarantee the synthetic data captures true market dynamics; high likelihood does not equal realistic market behaviour.

**Questions:**

(1) Can you add error bars or other measures to account for randomness in the cumulative return plots? Statistical significance testing (e.g., bootstrap confidence intervals, performance across different random seeds) would help distinguish genuine improvements from noise.

(2) I do not have sufficient information to reproduce your results. I could not find the code or a GitHub link (unless I missed it). If reproducibility issues are addressed, particularly by releasing code and detailed implementation details, I would consider raising my score.

---

### Note · Authors · 2025-11-20

I have read and agree with the venue's withdrawal policy on behalf of myself and my co-authors.